# Non-Immersive Virtual Reality for Rehabilitation of the Older People: A Systematic Review into Efficacy and Effectiveness

**DOI:** 10.3390/jcm8111882

**Published:** 2019-11-05

**Authors:** Roberta Bevilacqua, Elvira Maranesi, Giovanni Renato Riccardi, Valentina Di Donna, Paolo Pelliccioni, Riccardo Luzi, Fabrizia Lattanzio, Giuseppe Pelliccioni

**Affiliations:** 1Scientific Direction, IRCCS INRCA, 60129 Ancona, Italy; r.bevilacqua@inrca.it (R.B.); f.lattanzio@inrca.it (F.L.); 2Clinical Unit of Physical Rehabilitation, IRCCS INRCA, 60100 Ancona, Italy; g.riccardi@inrca.it (G.R.R.); v.didonna@inrca.it (V.D.D.); 3Eye Clinic, Polytechnic University of Marche, 60100 Ancona, Italy; paopel@hotmail.it; 4Medical Direction, IRCCS INRCA, 60100 Ancona, Italy; r.luzi@inrca.it; 5Neurology Unit, IRCCS INRCA, 60100 Ancona, Italy; g.pelliccioni@inrca.it

**Keywords:** virtual reality, cognitive and physical rehabilitation, oldest old person

## Abstract

Objective: the objective of this review is to analyze the advances in the field of rehabilitation through virtual reality, while taking into account non-immersive systems, as evidence have them shown to be highly accepted by older people, due to the lowest “cibersikness” symptomatology. Data sources: a systematic review of the literature was conducted in June 2019. The data were collected from Cochrane, Embase, Scopus, and PubMed databases, analyzing manuscripts and articles of the last 10 years. Study selection: we only included randomized controlled trials written in English aimed to study the use of the virtual reality in rehabilitation. We selected 10 studies, which were characterized by clinical heterogeneity. Data extraction: quality evaluation was performed based on the Physioterapy Evidence Database (PEDro) scale, suggested for evidence based review of stroke rehabilitation. Of 10 studies considered, eight were randomized controlled trials and the PEDro score ranged from four to a maximum of nine. Data synthesis: VR (Virtual Reality) creates artificial environments with the possibility of a patient interaction. This kind of experience leads to the development of cognitive and motor abilities, which usually positively affect the emotional state of the patient, increasing collaboration and compliance. Some recent studies have suggested that rehabilitation treatment interventions might be useful and effective in treating motor and cognitive symptoms in different neurological disorders, including traumatic brain injury, multiple sclerosis, and progressive supranuclear palsy. Conclusions: as it is shown by the numerous studies in the field, the application of VR has a positive impact on the rehabilitation of the most predominant geriatric syndromes. The level of realism of the virtual stimuli seems to have a crucial role in the training of cognitive abilities. Future research needs to improve study design by including larger samples, longitudinal designs, long term follow-ups, and different outcome measures, including functional and quality of life indexes, to better evaluate the clinical impact of this promising technology in healthy old subjects and in neurological patients.

## 1. Introduction

Virtual reality (VR) is a trending, widely accessible, contemporary technology of increasing utility to biomedical and health applications [1]. VR is the technological experience that allows for a full immersion in virtual spaces with which you can interact via specific wearable or using only your hand. A key feature of all VR applications is interaction. Virtual environments (VE) are created and allow for the user to interact with not only the VE, but also with virtual objects within the environment. In some systems, the interaction might be achieved via a pointer operated by a mouse or joystick button. In other systems, a representation of the user’s hand (or other body part) might be created within the environment where the virtual hand movement is generated [2].

VR ranges from non-immersive to fully immersive, depending on the degree to which the user is isolated from the physical surroundings when interacting with the virtual environment. Non-immersive virtual reality allows for interacting with the environment through mouse or joystick; immersive virtual reality, instead, uses tools that are connected to the human body in order to perform the same motor task [3,4]. Non-immersive VR systems have been studied as a therapeutic tool for improving symptoms in neurological disorders and have shown potential to promote cognitive and motor improvements even in advanced stages of different neurological diseases (e.g., stroke, Alzheimer and Parkinson disease (AD, PD), multiple sclerosis (MS), and traumatic brain injury) because of these characteristics [5,6,7,8,9].

The use of VR technology in rehabilitation derives from research in computational neuroscience involving motor learning mechanisms [10]. VR provides real-time visual feedback for movements, thereby increasing engagement in enjoyable rehabilitation tasks [11].

VR provides alternative rehabilitation programs with new and effective therapeutic tools that can improve the functional abilities in a wide variety of rehabilitation patients in a neurological setting, offering several features, such as goal-oriented tasks and repetition. The use of VR environments for virtual augmented exercise has recently been proposed as having the potential to increase exercise behavior in older adults [12] and it also has the potential to influence cognitive abilities in this population segment [13]. Therefore, VR represents a real opportunity for the cognitive rehabilitation of neurological patients with different neuropsychological symptoms, especially in attention, memory, problem-solving and executive dysfunction, and in behavioral impairments [7,8,9].

Moreover, VR training has been mostly described for the upper limb [14,15], but also for the lower limb [16], balance and walking [17,18], as well as for perceptual/cognitive skills [19].

To our knowledge, systematic reviews or meta-analyses have been undertaken to review the utility of VR technologies in a single arm of rehabilitation (i.e., motor or cognitive rehabilitation, upper or lower limb rehabilitation), focusing on a specific pathology (stroke, PD, AD, MS) [6,7,9].

Despite the growing evidence of the positive effects of VR in rehabilitation of functional and cognitive abilities, some systems still raised concerns regarding their acceptability with complex clinical populations, as, for example, the older people. In particular, during trials with immersive systems, few adverse events have been described by participants, including headache and dizziness [20]. Finally, little is known about the perceived effect of the exposure at multisensory input during a complex activity, such as treadmill walking with VR in patients during post-stroke rehabilitation to improve balance and gait ability [6,21].

The objective of this review is to analyze the advances in the field of rehabilitation through VR, while taking non-immersive systems into account, as evidence have shown to be highly accepted by older people, due to the lowest “cibersikness” symptomatology [20]. For this purpose, Randomized Controlled Trials (RCTs) were analyzed in order to investigate the effects of rehabilitation programs integrated with innovative non-immersive VR systems and suggest future clinical applications.

## 2. Methods

### 2.1. Literature Search and Study Selection

The methodology of this systematic review was based on the Preferred Reporting Items for Systematic Reviews and Meta-Analyses (PRISMA) guidelines, as the main aim of this work is mapping all the available literature in the rehabilitation with non-immersive virtual reality. A systematic review of the literature was conducted in June 2019. The data were collected from Cochrane, Embase, Scopus, PubMed, and Science Direct databases, analyzing manuscripts and articles of the last 10 years (from June 2009 to June 2019), in order to obtain the latest evidence in the field.

Based on consultation with the multidisciplinary team, non-immersive VR studies and applications related to rehabilitation intervention were searched while using the following search terms, and the combination thereof: non-immersive, virtual reality, virtual game, rehabilitation, motor impairment, and cognitive impairment.

After the preliminary search, 26 articles resulted from PubMed, 19 from Scopus, 283 from Science Direct, 10 from Embase, and 11 from Cochrane.

The findings were analyzed and screened by four experts of the team, a bioengineer, a clinical neuropsychologist, a statistician, and a neurologist. In particular, three review authors independently reviewed titles and abstracts that were retrieved from the search in order to determine whether they met the predefined inclusion criteria. A fourth review author (a statistician) moderate any disagreement. The full text articles were subsequently analyzed.

The first screening was based on the analysis of the title and of the abstract, as well as deduplication of the findings. Another researcher confirmed the accuracy of the papers selection and screened for any possible omission. After the first step, 11 articles resulted from PubMed, two from Scopus, and 0 from Science Direct, Embase, and Cochrane.

### 2.2. Study Selection

We included RCTs and reviews written in English that aimed to study the use of non-immersive virtual reality in rehabilitation. Thus, we selected studies meeting the following criteria:Studies conducted on adult patients aged ≥65 yearsStudies devoted to use a non-immersive virtual reality in rehabilitationStudies including upper limb rehabilitation, lower limb rehabilitation, or cognitive rehabilitationRandomized clinical trials, with control group that received conventional rehabilitation therapyBefore-after comparison of a single groupReview articles

On the contrary, we excluded studies that met the following criteria:Conference proceedingsStudies for which the full text was not foundStudies written in languages other than EnglishTechnical papersQualitative studies

All case-report studies and case-control studies were excluded for a lack of sustainability of results, as well as works concerning the development of new technologies.

### 2.3. Data Collection

After the screening based on the inclusion/exclusion criteria, conducted on the full text articles, the studies were selected as follows: 0 from Scopus, 10 from PubMed, and no one from Cochrane, Science Direct and Embase, and one from other sources. The countries of the selected studies are: Spain (2), France (2), Italy (2), Israel (1), United State of America (1), Canada (2), and Brazil (1). The fact that the studies have been performed in different countries shows that the topic is of general interest. Figure 1 shows the flowchart search strategy applied.

## 3. Results

A total of 10 papers were included. Studies are both reviews [22,23] and clinical papers [24,25,26,27,28,29,30,31].

### 3.1. Study Quality Evaluation

Quality evaluation was performed based on the PEDro scale and on the Cochrane’s Risk of Bias (RoB) tool, suggested for evidence based review of rehabilitation while using non-immersive virtual reality [32,33]. The final score was settled when three authors reached agreement after repeated review and analysis. Of eight studies considered, five were randomized controlled trials and the PEDro score ranged from four to a maximum of nine, and the RoB score ranges from one to five (Table 1).

### 3.2. General Characteristics of the Study Population

All of the studies were focused on older people with a mean age of 65.2 (±9.4) years for the experimental group and 69.2 (±9.4) years in the control group. The number of participants that were involved in all the studies is 1008 ranged from 6 to 376.

To our knowledge, of the older people involved in the trials, 586 were males and 392 females.

The majority of the patients suffered from stroke (*n* = 593), followed by older people at high risk of falls, with more than two falls in six months (*n* = 182), patients with amyotrophic lateral sclerosis (*n* = 30), AD (*n* = 24), or PD (*n* = 24).

### 3.3. Descriptive Analysis and Outcome Measures

Table 2 shows the characteristics of the studies. The outcome could not be pooled into meta-analysis due to the following reasons. Clinical heterogeneity (Table 2) can be clearly observed from the participant, intervention, exercise mode, and outcome measures of the included studies. Diversity is seen in patient conditions and pathology, frequency, and duration of VR intervention, whether the impairment concerns the upper or the lower limb, whether the experiment conducted was pure VR (only VR) or VR mixed with traditional physical therapy or with exercise therapy, and whether the outcome measure contains follow-up.

### 3.4. Intervention Effects

Eight papers report the results of clinical trials involving a group of patients that performed a training with VR system versus a control group that performed a traditional physiotherapy training [25,26,27,28,29,31], or a comparison within the same group performing a VR training while using different exercise control modalities [30].

The period of VR training ranged from four to six weeks, each day of the week or three training sessions per week, while the duration of each single session with VR system ranged from 15 to 60 minutes. Only in one study [27] the duration of intervention was two weeks. Generally, all the experimental groups (EGs) in the studies have received both therapies with VR and traditional physiotherapy, while the control groups (CGs) have only received traditional physiotherapy. Two studies have a follow up after eight and 12 weeks [29] or six months [28].

The study of Walker et al. [24] reported the lowest number of subjects (*n* = 6) within one-year post-stroke. It is a before-after study and all the subjects performed training with a treadmill equipped with a VR system. All the participants made significant improvements in their ability to walk, increasing the over ground walking speed and the Berg Balance Scale (BBS) scores.

The study of Turolla et al. [25] involves 376 post-stroke patients randomized into two groups, receiving combined VR and upper limb conventional therapy or traditional therapy alone. VR rehabilitation seems more effective than conventional interventions in restoring upper limb motor impairments and motor related functional abilities.

The study of Allain et al. [26] involves 24 Alzheimer’s disease patients as compared with 32 healthy elderly controls on a task designed to assess their ability to prepare a virtual cup of coffee, comparing the performance with an identical daily living task. Significant relations are found between virtual and real coffee-making scores, and between virtual score and Instrumental activities of daily living (IADL) scale, which supports the validity of the virtual reality training.

In a study of Saposnik et al. [27], 141 post-stroke patients were randomized into two groups: the first received the VR therapy and the second received recreational therapy. The results show that within each group the performance time improves from baseline to the end of treatment, whereas no differences are found between groups.

The objective of Mirelman et al. [28] was to verify whether an intervention combining treadmill training with non-immersive virtual reality (to target both cognitive aspects and mobility) would lead to fewer falls than treadmill training alone would. To do this, the authors recruited 282 older people at high risk of falls and randomized them into two groups to receive treadmill training plus VR or treadmill training alone. In the six months after training, the incident rate was significantly lower in the experimental group.

The study of Segura-Ortì et al. [29] involves 18 patients on hemodialysis: nine performed 30 minutes of non-immersive virtual reality training and nine performed 30 minutes of aerobic training. Both interventions improved physical function, such as gait speed and no significant differences, were found between groups.

In the study of Trevizan et al. [30], the performance on a computer task in patients with amyotrophic lateral sclerosis while using three (motion tracking, finger motion control, or touch screen) different commonly used non-immersive devices was evaluated. The control and experimental group both showed better performance on the computer task when using the touch screen device.

Pelosin et al. [31] analyzed 39 patients with Parkinson’s disease, assigned to treadmill training group or treadmill training with non-immersive virtual reality intervention group to assess cholinergic activity. The results showed that the experimental group improved obstacle negotiation performance, and reduced the number of falls as compared with control group.

## 4. Discussion

A review of the evidence on VR efficacy in patients affected by a neurological disease is mandatory due to the rapid development of VR programs in the last years and the increasing literature on VR application in neurological conditions, in order to enable clinicians to have an up-to-date understanding of the potential clinical beneficial effects of these techniques.

Therefore, the aim of this paper was to systematically evaluate the evidence of the effectiveness of VR compared to conventional therapies. It must be stressed that few studies summarize the current best evidence on the effectiveness, user compliance, feasibility, and safety of VR interventions for rehabilitation treatment in neurological disorders.

VR creates artificial environments with the possibility of a patient interaction. This kind of experience leads to the development of cognitive and motor abilities, which usually positively affect the emotional state of the patient, increasing collaboration and compliance.

Moreover, the VR rehabilitative treatment might be personalized according to the specific abilities and needs of the subject.

Parkinson disease is one of the most common age-related brain disorders with both dopamine-related motor symptoms and nonmotor symptoms due to other neurotransmitter circuits involvement, such as the cholinergic, noradrenergic, and serotonergic pathways.

The cognitive decline is among the most common and relevant nonmotor symptoms in PD and it affects different cognitive domains, in particular attentional, visuospatial and executive domains, and also memory. VR in the cognitive PD treatment could be useful in improving, in particular, the visuospatial and executive abilities, which represent the most compromised aspects of cognitive decline in PD patients [7].

Moreover, falls are frequent in ageing and PD patients, due to an impairment in the cholinergic-mediated gait pathway. A rehabilitation approach using treadmill training combined with non-immersive VR seems to induce changes in cortical cholinergic activity, which enables functional gait improvements and reduces the fall rate in comparison to a traditional rehabilitation method [31].

A reduction in static and dynamic balance is a major risk factor for falls also in stroke survivors [34]. In fact, the majority of the individuals with stroke who have fallen usually develop fear of falling again (88%). Fear of falling is related to balance and gait deficits [35], and it often leads to reduced physical activity and deconditioning. In fact, 44% of stroke fallers report restriction of activity after the fall. Given the very low physical activity and cardiovascular fitness levels already near the lower limit of those required for basic ADL, further activity reduction and deconditioning due to the fear of falling can easily lead to a loss of independence in individuals with stroke.

A recent review on post-stroke rehabilitation therapy [6] provided evidence for a moderate beneficial effect in balance improvement of VR combined with conventional therapy, as compared to conventional therapy alone.

More promising effects seem to be evident in the case of upper limb motor impairments in stroke rehabilitation [25], but further studies are needed on this subject. In fact, the trial described in [27] found that non-immersive VR as an add-on therapy to conventional rehabilitation was not superior to a recreational activity intervention in improving motor function, which suggested that the added intensity of training only induces early motor recovery of the upper limb, and that this can be achieved with VR or with other simple and inexpensive arm activities.

VR technology has considerable potential for detecting functional limitations in IADL performance in AD patients, beyond that of current neuropsychological measures, as shown by Allain et al. [26]. Moreover, studies were carried out to assess the effectiveness of a VR cognitive training program on cognition in mild cognitive impairment (MCI) and AD patients [36].

VR cognitive training for individuals with MCI and dementia has proven to result in improvements in the cognitive domains of attention, executive function, and visual and verbal memory. Moreover, significant reductions in depressive symptoms and anxiety were evident, with a delay in the progression of cognitive impairment [37].

Additionally, the VR format might help in training adherence, as individuals with MCI and dementia patients seem to prefer the VR format of a task over the paper version, as confirmed by a feasibility study with image-based rendered VR in patients with mild cognitive impairment and dementia [38].

Some recent studies have suggested that rehabilitation treatment interventions might be useful and effective in treating motor and cognitive symptoms in different neurological disorders, including traumatic brain injury [8], multiple sclerosis [9], and progressive supranuclear palsy [39].

Finally, VR represents an effective tool that could improve the traditional cognitive and motor rehabilitation in patients that are affected by a neurological disease. Moreover, home-based VR might offer a promising addition or alternative to existing rehabilitation programs, and a chance to provide and/or prolong the required therapy after discharge in a more accessible setting, potentially improving clinical outcomes.

Future research needs to improve the study design by including larger samples, longitudinal designs, long term follow-ups, and different outcome measures, including functional and quality of life indexes, to better evaluate the clinical impact of this promising technology in healthy old subjects and in neurological patients. In particular, the next challenge for the research on VR and rehabilitation can be summarized in the following questions:

1. Does an innovative intervention enriched with VR provide a significant improvement in mobility, compared to traditional physiotherapy?

2. Is the intervention cost-effective for the health management systems?

To answer to these questions, it is crucial to understand how to improve the rehabilitation path of older people, through multidisciplinary multicomponent and person-centered intervention, integrated with VR.

The evidence reported in the paper are in line with the aims that were expressed by the National Plan for Health Research, whose priorities are defined in accordance with the indications contained in the regulation of the European Parliament and of the Council on the establishment of the “Health for growth” program, which pursues as a goal the achievement of a strong potential for economic growth thanks to the improvement of the state of health, through the facilitation of innovation in health care, the improvement of skills and information on specific diseases, and the identification of good practices for effective prevention [40]. In line with what has been expressed, the role of health technology assessment is of crucial importance. In this perspective, the available services must necessarily be enriched with adequate equipment of proven efficacy, as the promising sector of VR, to be able to advance at both the methodological and assistance level.

## 5. Conclusions

As it is shown by the numerous studies in the field, the application of VR has a positive impact on the rehabilitation of the most predominant geriatric syndromes. The level of realism of the virtual stimuli seems to have a crucial role in the training of the cognitive abilities. Nevertheless, semi-immersive or non-immersive VR systems have the advantage of being more accepted by the users, as they experienced less cybersickness after the training. Moreover, the integration of these devices in the health management systems are still lacking despite the evidence and the peculiarity of VR technologies with different level of immersivity. A tentative explanation can be found, not only in the cost of technology that seems to be more affordable in the recent years, but, most of all, in the absence of a standardized protocol and procedure, to harmonize traditional rehabilitation therapies and innovative VR systems. For this reason, it will be necessary to improve the research in the field, adopting RCTs study design as well as indicators of health technology assessment, to understand the effectiveness and efficacy also in terms of optimization of the clinical pathways. In addition, as VR systems can be easily adopted at home, it can be considered to be useful for the continuity of care.

## Figures and Tables

**Figure 1 jcm-08-01882-f001:**
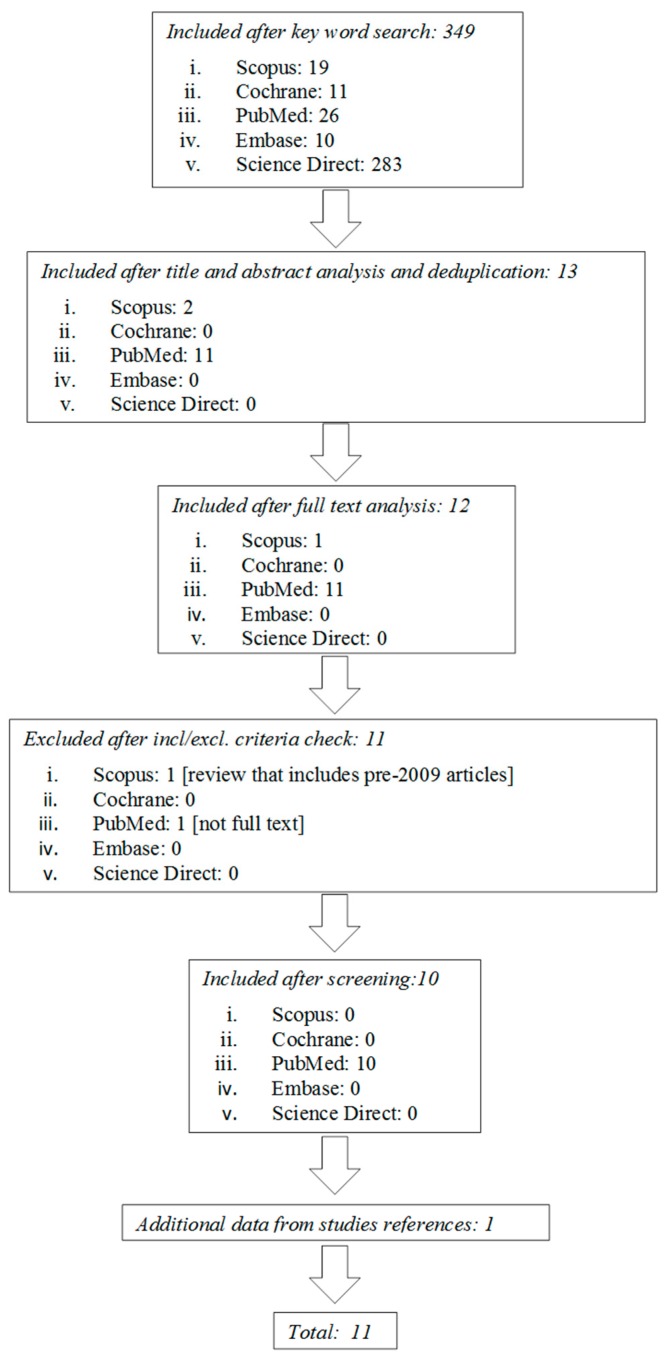
Flow diagram of the study selection process.

**Table 1 jcm-08-01882-t001:** Scores of methodological quality assessment of the included studies.

PEDro [32]	Walker et al., 2010 [24] RCT	Turolla et al., 2013 [25] RCT	Allain et al., 2014 [26] RCT	Saposnik et al., 2016 [27] RCT	Mirelman et al., 2016 [28] RCT	Seguera-Ortì et al., 2018 [29] RCT	Trevizan et al., 2018 [30] RCT	Pelosin et al., 2019 [31] RCT
Eligibility	Y	Y	Y	Y	Y	Y	Y	Y
Randomized allocation	N	N	N	Y	Y	Y	Y	Y
Concealed allocation	N	N	N	Y	Y	Y	Y	Y
Baseline comparability	Y	Y	N	Y	N	N	N	N
Blinded subject	N	N	N	N	N	N	N	N
Blinded therapists	N	N	N	N	N	N	N	N
Blinded raters	N	N	N	Y	Y	Y	Y	Y
Key outcomes	Y	Y	Y	Y	Y	Y	Y	Y
Intention to treat	N	N	N	Y	Y	Y	Y	N
Comparison between groups	N	Y	Y	Y	Y	Y	Y	Y
Precision and variability	Y	Y	Y	Y	Y	Y	Y	Y
	4/11	5/11	4/11	9/11	8/11	8/11	8/11	7/11
**Cochrane’s Risk of bias tool** [33]								
Sequence generation	N	N	N	Y	Y	Y	Y	Y
Allocation concealment	N	N	N	Y	Y	Y	Y	Y
Blinding of participants, personnel and outcome assessors.	N N N	N N N	N N N	N N Y	N N Y	N N Y	N N Y	N N Y
Incomplete outcome data.	N	N	N	N	N	N	N	N
Selective outcome reporting	Y	Y	Y	Y	Y	Y	Y	Y
Other sources of bias	N	Y	Y	Y	Y	N	N	Y
	1/8	2/8	2/8	5/8	5/8	4/8	4/8	5/8

Y: yes; N: no.

**Table 2 jcm-08-01882-t002:** Descriptive analysis of the included clinical studies.

	Population		Technological Devices	Intervention		Comparison	Outcome
	Partecipants in Experimental Group	Partecipants in Control Group		Training Frequency	Intervention Group	Control Group	
Walker et al., 2010 [24]	6 adults within 1-year post-stroke *N* = 6, 3 F/3 M Age: 54.3 years (range 41–70 years)	-	A partial body weight-support treadmill in conjunction with a television mounted on a stand in front of the treadmill to display the VR walkthrough environment.	2 or 3 training sessions per week with partial body weight-supported tredmill with virtual reality system (total 12 training sessions). Initial training duration is 10 minutes; duration was progressed as tolerated.	-		(1) FGA scores increased by 30% (2) BBS scores improved by 10% (3) Overground walking speed increased by 38%
Turolla et al., 2013 [25]	*n* = 263 post-stroke patients, 105 F/158 M Age: 60.2 ± 14.3 years	*n* = 113 post-stroke patients, 41 F/72 M Age: 65.4 ±12.5	The Virtual Reality Rehabilitation System (Khymeia group. Noventa padovana, Italy) includes a pc workstation connected to 3D motion-tracking system and a high-resolution LCD projector displaying the virtual scenarios on a large wall screen.	40 sessions of daily therapy provided 5 days per week, for 4 weeks.	40 sessions of daily therapy provided 5 days per week, for 4 weeks. 1 hour of conventional therapy and 1 hour of VR therapy	2 hours of conventional treatment.	Within groups: F-M UE score improved by 4% in control group, and 10% in experimental group. Between groups: significantly greater motor improvement in experimental group. Within groups: FIM scores improved in both groups. Between groups: FIM scores improved by 5% in experimental group than in control group.
Allain et al., 2014 [26]	*n* = 24 AD patients, 14 F/10 M Age: 76.96 ± 6.05 years	*n* = 32 healthy older patients, 25 F/7 M Age: 74.13 ± 5.93	The virtual environment simulated a fully texture, medium-size kitchen. In the foreground, there was a work plane with all the objects needed to prepare a cup of coffee with milk and sugar. Patients controlled the 2D cursor using a computer mouse.	1. Virtual reality: 3 sessions: 2 of training and one test session to prepare a cup of coffee in virtual condition 2. Reality: to prepare a cup of coffee	Each training sessions lasts 15 minutes	Each training sessions lasts 15 minutes	Within groups: time to complete the virtual task and MMSE score are correlated in both groups
Saposnik et al., 2016 [27]	*n* = 71 stroke patients, 25 F/46 M Age: 62 ± 13 years	*n* = 70 stroke patients, 22 F/48 M Age: 62 ± 12 years	The Nintendo Wii gaming system or recreational activities (playing cards, bingo, jenga or ball game).	10 sessions, 60 minutes each, over a 2 week period.	30 minutes of traditional rehabilitation of the upper extremity + 30 minutes of virtual reality training	60 minutes of traditional rehabilitation of the upper extremity	1. Within groups: WMFT performance time improves from baseline to the end of treatment in both groups. 2. Between groups: no differences in WMFT at the end and at 4-weeks post-intervention 3. Between groups: better performance in BBT in control group at the end of treatment.
Mirelman et al., 2016 [28]	*n* = 146 older people at high risk of falls (more than 2 falls in 6 months), 48 F/98 M Age: 74.2 ± 6.9 years	*n* = 136 older people at high risk of falls (more than 2 falls in 6 months), 52 F/84 M Age: 73.3 ± 6.4 years	The treadmill plus VR intervention included a camera for motion capture and a computer generated simulation. The virtual environments included real-life challenges with obstacles, multiple pathway and distracters.	3 times per week for 6 weeks, with each session lasting about 45 minutes	45 minutes of treadmill training with virtual reality	45 minutes of traditional treadmill training	In the 6 months after training, the incident rate was significantly lower in the treadmill training plus VR group.
Seguera-Ortì et al., 2018 [29]	*n* = 9 patients on hemodialysis, 4 F/5 M Age: 61.8 ± 13.0	*n* = 9 patients on hemodialysis, 3 F/6 M Age: 68.3 ± 15.6	The system is an adapted version of ACT (A la Caza del Tesoro), in which the subject tries to catch a series of targets by moving their leg.	16 weeks of intra-dialysis exercise program. The program lasted 4 additional weeks.	5 minutes warm-up; 30 minutes of virtual reality training.	5 minutes warm-up and strengthening exercises; 30 minute of aerobic training; 5 minutes of stretching.	1. Between groups: no significant differences in STS-60 2. Between groups: significant differences in gait speed 3. Within groups: significant improvements for STS-10, gait speed, 6 minute walking test between baseline-16 and 20 weeks, and 16–20 weeks.
Trevizan et al., 2018 [30]	*n* = 30 people with ALS, 12 F/18 M Age: 59 years (range 44–74 years)	*n* = 30 healthy people, equally matched for age and gender with experimental group	The VR environment is a 3D game in which the goal was to reach as many bubbles displayed on the computer monitor. The game was controlled by three different device system: motion tracking, finger motion and touch-screen.	Participants were randomly divided in 3 groups: motion tracking, finger motion control, touchscreen, to perform 3 task phases (acquisition, retention, transfer)			Both experimental and control group showed better performance whn using the touchscreen device in the transfer phase.
Pelosi net al., 2019 [31]	*n* = 10 PD + 7 OA 11 F/6 M Age: 73.2 ± 3.6	*n* = 14 PD + 8 OA 15 F/7 M Age: 71.9 ± 4.1	Treadmill with a non-immersive virtual reality that reacts to a virtual environment that included real-life challenges	45 minutes/session, 3 times a week for 6 weeks	To walk on a treadmill with virtual reality that included obstacles, distracters.	To walk on a treadmill without virtual reality.	Experimental group increased SAI, reduced the number of falls, improved obstacle negotiation performance.

FGA: Functional Gait Assessment. BBS: Berg Balance Scale. VR: Virtual Reality. F-M UE: Fugl-Meyer upper extremity. FIM: Functional Independence Measure. AD: Alzheimer’s disease. MMSE: Mini Mental State Examination. WMFT: Wolf Motor Function Test. BBT: Box and Block Test. STS-60: sit-to-stand tests 60. ARAT: Action Research Arm Test. MMAS: Modified Modified Ashworth Scale. MAL: Motor Activity Log. FSS-7: Fatigue Severity Scale seven-item. SIPSO: Subjective Index of Physical and Social Outcome 10-item. VAS: Visual Analogue Scale. ALS: Amyotrophic Lateral Sclerosis. PD: Parkinson Disease. OA: older adults. SAI: Short-latency afferent inhibition.

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
