# Peer review of "Non-Immersive Virtual Reality for Rehabilitation of the Older People: A Systematic Review into Efficacy and Effectiveness"

_jcm, 2019, doi:10.3390/jcm8111882_

Round 1
Reviewer 1 Report
In this paper the authors have analyzed the use of Virtual Reality in rehabilitation processes with different pathologies such as Stroke, Amyotrophic Lateral Sclerosis, Parkinson’s disease, patients with Hemodialysis, and Alzheimer’s disease. For this purpose, they collected information in different electronic databases and selected only papers with randomized control trials. They evaluated the quality of the selected papers by using PEDRO scale and finally the concluded that the use of VR alleviates motor alterations in multiples pathologies and helps in training sessions of the cognitive abilities.
This paper deals with an area of research that is important, with a reasonable standard, however it does contain flaws.
The first point is related to the methodology section:
The authors need to extend their search with other electronic databases such as Pubmed, Ieee Xplore, or DARE.
The authors need to describe more the typical terms that they used. “non-immersive AND virtual reality AND rehabilitation”?
The authors need to include a paragraph describing the countries of the selected studies.
They need to describe the stages to select the articles by the four experts: evaluation of the title of the papers, reading the abstract, and finally assessment of the full paper.
Another important point in the paper is the analysis of risk of bias: selection, performance, detection, attrition, and reporting. The authors need to analyze the quality and the risk of biases in selected studies.
In Table 2 the authors need to put two columns indicating chronicity of patients and technological devices used in the studies.
In discussion Section, authors describe “A reduction in static and dynamic balance is a major risk factor for falls also in stroke survivors”, they need to discuss this phrase with a reference.
Minor Compulsory Revisions –
There are the word virtual reality in multiples sections on the paper, please modify by VR.
Level of interest: this is an article of importance in its field.
The quality of written English is acceptable.
Author Response
We would like to thank the reviewer for his/her helpful comments and suggestions. We very much appreciate the time and effort invested in the review of our manuscript. We carefully scrutinized our original paper and considered each of the comments. An itemized list of the comments and our response and changes are listed below. We hope that you will find our paper, in its present form, acceptable for publication in Journal of Clinical Medicine.
Reviewer’s comments:
In this paper the authors have analyzed the use of Virtual Reality in rehabilitation processes with different pathologies such as Stroke, Amyotrophic Lateral Sclerosis, Parkinson’s disease, patients with Hemodialysis, and Alzheimer’s disease. For this purpose, they collected information in different electronic databases and selected only papers with randomized control trials. They evaluated the quality of the selected papers by using PEDRO scale and finally the concluded that the use of VR alleviates motor alterations in multiples pathologies and helps in training sessions of the cognitive abilities.
This paper deals with an area of research that is important, with a reasonable standard, however it does contain flaws.
The first point is related to the methodology section:
C1: The authors need to extend their search with other electronic databases such as Pubmed, Ieee Xplore, or DARE.
A1: Thank you for the suggestion. Nevertheless, Pubmed was already included in our research (Abstract line 10; Methods line 98) while IEEE Xplore had not been quoted because, despite having consulted it, it gathered only conference papers, more technical than clinical.
C2: The authors need to describe more the typical terms that they used. “non-immersive AND virtual reality AND rehabilitation”?
A2: Thank you for the comment. The sentence has been rephrased and the search strategy has been clarified (lines 100-103).
C3: The authors need to include a paragraph describing the countries of the selected studies.
A3: Thank you for the suggestion. We included this information in the 2.3 section (Data collection), lines 137-140.
C4: They need to describe the stages to select the articles by the four experts: evaluation of the title of the papers, reading the abstract, and finally assessment of the full paper.
A4: Thank you for this comment. The stage to select the articles have been described in more detail in the 2.1 section (lines 106-110)
C5: Another important point in the paper is the analysis of risk of bias: selection, performance, detection, attrition, and reporting. The authors need to analyze the quality and the risk of biases in selected studies.
A5: Thank you for the suggestion. In the revised manuscript we have added the Cochrane’s Risk of Bias (RoB) tool, both in the text (lines 144-148, reference 34) and in the Table 1.
C6: In Table 2 the authors need to put two columns indicating chronicity of patients and technological devices used in the studies.
A6: These information has been added in the table 2: the chronicity in the population column and the technological devices used inserted a column, highlighted in yellow.
C7: In discussion Section, authors describe “A reduction in static and dynamic balance is a major risk factor for falls also in stroke survivors”, they need to discuss this phrase with a reference.
A7: This phase has been discussed in the discussion session (lines 249-254) and the references have been updated (ref. 35 and 36, highlighted in yellow in the references list).
Minor Compulsory Revisions –
C8: There are the word virtual reality in multiples sections on the paper, please modify by VR.
A8: Done. Thank you.
Please see the attachment (manuscript modified in revision modality)

Reviewer 2 Report
The authors present a very interesting review of the state-of-the-art on non immersive virtual reality approaches for rehabilitation of older people. The authors researched and found 11 approaches regarding this field. The paper is easy to read and insightful. However there are some open topics that I think would improve the quality of the paper.
First, given the short amount of papers found, I wonder if it would not be best for the authors to broaden a bit their scope to non-immersive VR for rehabilitation without an age restriction, or even to e-health in general. I know there are way more papers regarding other fields of e-health, such as exposure therapy.
Second, the location of Section 3 is very weird with the page in a completely different direction. I wonder if it would be possible to structure it differently (the table could take the full page).
Third, the discussion section is seems to be a simple summary, without any insights. What are the open challenges to be addressed on non-immersive VR for rehabilitation? This should be addressed.
Author Response
We would like to thank the reviewer for his/her helpful comments and suggestions. We very much appreciate the time and effort invested in the review of our manuscript. We carefully scrutinized our original paper and considered each of the comments. An itemized list of the comments and our response and changes are listed below. We hope that you will find our paper, in its present form, acceptable for publication in Journal of Clinical Medicine.
Reviewer’s comments:
The authors present a very interesting review of the state-of-the-art on non immersive virtual reality approaches for rehabilitation of older people. The authors researched and found 11 approaches regarding this field. The paper is easy to read and insightful. However, there are some open topics that I think would improve the quality of the paper.
C1: First, given the short amount of papers found, I wonder if it would not be best for the authors to broaden a bit their scope to non-immersive VR for rehabilitation without an age restriction, or even to e-health in general. I know there are way more papers regarding other fields of e-health, such as exposure therapy.
A1: Thank you for the suggestion. Unfortunately, the aim of this review is precisely to analyze the use of non-immersive virtual reality in the rehabilitative intervention in the older people. Expanding the research, we would seem to distort the final goal of the paper. The mission of our Institution is to focus on ageing and age-related diseases, so the topic of the paper is to understand how VR can be properly used in this population. The availability of 11 studies suggests the urgent need of a deeper investigation both in terms of achievement of background knowledge and conduction of RCT, aimed at the assessment of health outcomes. This review should constitute a sort of starting point for innovative clinical intervention.
C2: Second, the location of Section 3 is very weird with the page in a completely different direction. I wonder if it would be possible to structure it differently (the table could take the full page).
A2: In our opinion, due to the size of the table, inserting it in the horizontal direction is the only way to make it readable.
C3: Third, the discussion section is seeming to be a simple summary, without any insights. What are the open challenges to be addressed on non-immersive VR for rehabilitation? This should be addressed.
A3: Thank you for the comments. We have added some further information on future challenges for VR in rehabilitation setting. Lines 287-304.
Please see the attachment (manuscript modified in revision modality).

Round 2
Reviewer 1 Report
This paper is well organized, with a material appropriate for the journal, with a quality of written English acceptable, and without any mayor/minor to make.
So, my final recommendation is to accept, published as “Review Article”
Reviewer 2 Report
The authors have addressed my comments.